# Clusters of synaptic inputs on dendrites of layer 5 pyramidal cells in mouse visual cortex

Onur Gökçe[†], Tobias Bonhoeffer*, Volker Scheuss*

Department Synapses - Circuits - Plasticity, Max Planck Institute of Neurobiology, Martinsried, Germany

**Abstract** The spatial organization of synaptic inputs on the dendritic tree of cortical neurons plays a major role for dendritic integration and neural computations, yet, remarkably little is known about it. We mapped the spatial organization of glutamatergic synapses between layer 5 pyramidal cells by combining optogenetics and 2-photon calcium imaging in mouse neocortical slices. To mathematically characterize the organization of inputs we developed an approach based on combinatorial analysis of the likelihoods of specific synapse arrangements. We found that the synapses of intralaminar inputs form clusters on the basal dendrites of layer 5 pyramidal cells. These clusters contain 4 to 14 synapses within $\leq 30$ µm of dendrite. According to the spatiotemporal characteristics of synaptic summation, these numbers suggest that there will be non-linear dendritic integration of synaptic inputs during synchronous activation.

*For correspondence: tobias. bonhoeffer@neuro.mpg.de (TB); scheuss@neuro.mpg.de (VS)

Present address: [†]IBM Research – Zurich, Rüschlikon, Switzerland

Competing interests: The authors declare that no competing interests exist.

## Introduction

Computations on the level of single neurons critically depend on the location and spatial relationship between active synapses (e.g. *Branco and Häusser, 2011*; *Losonczy and Magee, 2006*; *Polsky et al., 2004*). Thus knowing the connectivity of neurons with single synapse resolution is essential to understand how synaptic integration at the cellular level contributes to neural circuit function. One type of spatial organization is of particular interest. These are clusters of synaptic inputs, which can give rise to superlinear summation during synchronous activity (e.g. *DeBello et al., 2014*; *Larkum and Nevian, 2008*; *Mel, 1993*). Clustering has been reported for synchronously active synapses during spontaneous activity (*Kleindienst et al., 2011*; *Takahashi et al., 2012*), for synapses undergoing plasticity (*Makino and Malinow, 2011*; *McBride et al., 2008*) as well as for synapses from anatomically defined populations of presynaptic neurons (*Druckmann et al., 2014*; *Rah et al., 2013*). However, the cluster parameters relevant for dendritic computations are not rigorously and quantitatively assessed. Furthermore, in different cortical areas the patterns of synapses activated by sensory stimulation suggest that inputs onto central neurons may not be clustered (*Chen et al., 2011*; *Jia et al., 2010*; *Varga et al., 2011*). Here we address two issues: First, we explore the spatial arrangement of synapses of an anatomically defined intracortical connection, i.e. the intralaminar connection between layer 5 pyramidal neurons. Secondly, we analyze the spatial organization of synapses with a novel method for identifying individual synapse clusters. This allows characterizing cluster parameters, such as dendritic length and number of inputs, in order to put them into the perspective of the rules of dendritic integration and the arithmetic of synaptic summation (e.g. *Branco and Häusser, 2011*; *Losonczy and Magee, 2006*; *Polsky et al., 2004*).

**eLife digest** Neurons in the brain exchange information through points of contact called synapses. If electrical activity arriving at a number of synapses exceeds a certain threshold, it can trigger an electrical impulse, which is transmitted to the next neuron. Synapses generally connect with branch-like structures called dendrites on the receiving neuron. However, little is known about how synapses are arranged on dendrites.

Gökçe et al. have now used a technique called optogenetics to work out the exact arrangement of a type of synapse on neurons in a part of the mouse brain that is devoted to vision. Optogenetics takes advantage of light-activated proteins that can trigger electrical activity. Gökçe et al. used mice that had been genetically engineered to produce these proteins in specific neurons, and then deliberately triggered electrical activity simply by shining light on these neurons. The experiments also used another technique called two-photon calcium imaging to monitor the activity of single synapses in response to the electrical activity triggered by optogenetics.

Gökçe et al. found that these neurons have clusters of four to fourteen synapses within a space of 30 micrometers along a dendrite. Synapses in clusters that are active at the same time can interact and thereby generate electrical signals more effectively than synapses spread across the dendrites. Further experiments are now needed to map the synapses between other kinds of neurons, and to map synapses from two different inputs at the same time.

## Results

For mapping the synapses between layer 5 (L5) pyramidal cells in mouse primary visual cortex (V1) we combined optogenetics and 2-photon calcium imaging (*Little and Carter, 2012*; *MacAskill et al., 2012*), which we refer to as *Channelrhodopsin (ChR2)-assisted synapse identity mapping* (CASIM). The principle of CASIM is depicted in *Figure 1A*: NMDA receptor mediated $Ca^{2+}$ signals identify those dendritic spines on a postsynaptic neuron, which receive input from photostimulated presynaptic neurons expressing ChR2. CASIM has the advantage over other methods for mapping synapses by light or electron microscopy (e.g. *Druckmann et al., 2014*; *Rah et al., 2013*) that it identifies synapses in functional rather than only in structural or molecular terms. For presynaptic expression of ChR2, we used transgenic mice expressing ChR2-YFP under control of the THY1 promoter in L5 pyramidal cells (line 18; *Wang et al., 2007*; *Figure 1B*). Expression levels and stimulus sensitivity were different from cell to cell. High photostimulus intensities evoked electrical responses in all probed L5 pyramidal cells. Because in more sensitive cells the number of generated action potentials dropped with higher stimulus intensities, we applied an intermediate stimulus intensity for synapse mapping, which evoked action potentials in 58% of L5 pyramidal cells (*Figure 1—figure supplement 1,2*). For synapse mapping we selected layer 5 pyramidal cells with no detectable ChR2-YFP fluorescence, indicating that the levels of ChR2 expression are so low that the selected stimulus intensity is insufficient to cause significant depolarization. For recording calcium signals in spines of postsynaptic L5 pyramidal cells, we patched and filled cells with Alexa 594 (30 µM) and Fluo-5F (1 mM) in acute slices. In order to isolate NMDA receptor mediated calcium signals and prevent spreading of unspecific excitation, patched cells were depolarized to 10 mV above the NMDAR reversal potential in voltage clamp in extracellular solution containing NBQX (10 µM), picrotoxin (50 µM) and D-serine (10 µM). Synapses between pairs of L5 pyramidal cells are reported to be located to a large extend on the proximal basal dendrite of the postsynaptic cell (*Markram et al., 1997*; *Sjoström and Häusser, 2006*). Therefore we focused on the basal dendrites of L5 pyramidal cells to map their inputs from other L5 pyramidal cells. Basal dendritic branches were systematically scanned with overlapping rectangular tiles of 2-photon calcium imaging to probe all spines for photostimulus-evoked calcium signals. *Figure 1C* shows a sequence of calcium imaging frames zoomed in on two spines, where the left spine shows a clear synaptic calcium signal. Due to the stochastic nature of neurotransmitter release, it is not expected that each AP leads to transmitter release. Thus we classified spines as receiving L5 input if they displayed at least once a higher ($\geq 3\times$ standard deviation) and earlier peak calcium signal evoked by photostimulation than observed in the parent dendrite (*Figure 1D - 1G*). In *Figure 1E* L5 input positive (L5+) spines are marked by white arrowheads

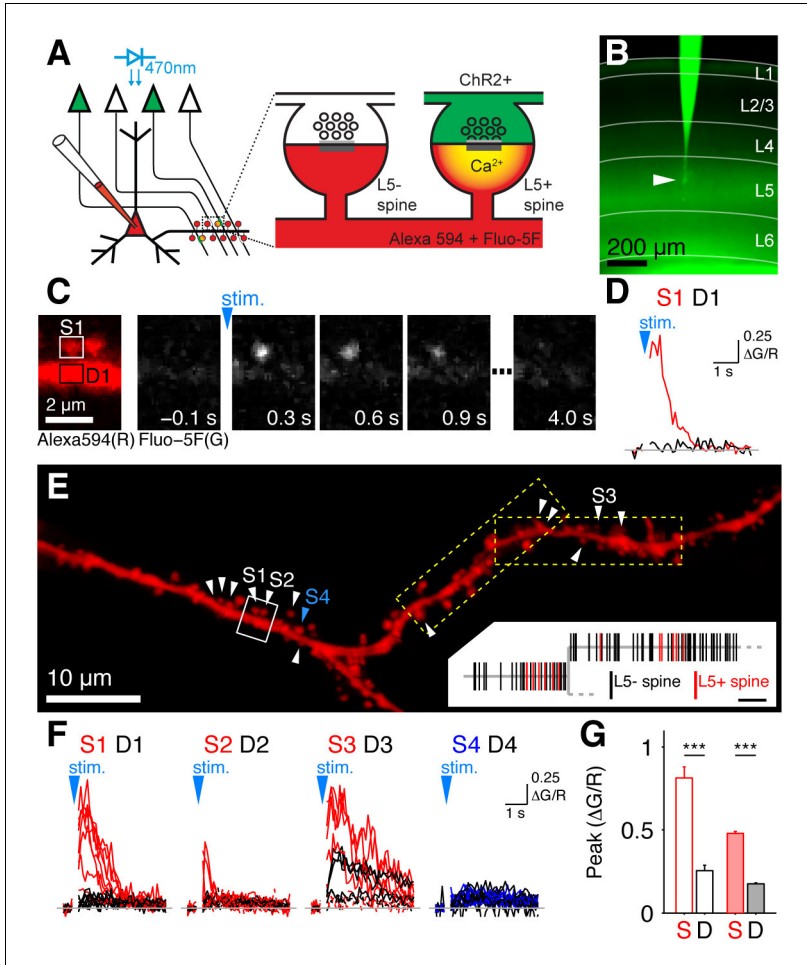

**Figure 1.** Mapping intralaminar inputs on the dendrites of L5 pyramidal neurons with channelrhodopsin-assisted synapse identity mapping (CASIM). (**A**) Schematic diagram illustrating the principle of CASIM. Synapses between presynaptic L5 neurons expressing ChR2 (green) and a postsynaptic L5 neuron filled with calcium indicator (red) are identified by calcium signals in dendritic spines (L5+ spines) evoked by photostimulation of ChR2. (**B**) Acute slice of primary visual cortex from a Thy1-ChR2 mouse. A L5 pyramidal cell (arrowhead) was patched and filled with Fluo-5F. (**C**) Example of a spine calcium signal evoked by ChR2 activation: Left, image of two spines and their parent dendrite (from white box in panel E) in the red channel (Alexa 594) for structural imaging. Right, sequence of frames from calcium imaging in one trial showing calcium signal in the left spine. (**D**) $Ca^{2+}$ signal corresponding to panel C ($\Delta G/R$; G, Fluo-5F; R, Alexa 594; red, spine; black, dendrite at spine base). (**E**) Analyzed dendritic stretch from cell in B. L5+ spines are marked by white arrowheads. Blue arrow head marks an example L5- spine. Dashed boxes indicate the size of imaging tiles. Inset, corresponding dendrogram; scale bar, 10 μm. (**F**) $Ca^{2+}$ signals from all trials recorded in a selection of L5+ spines (S1 to S3), one L5- spine (S4) and at their respective dendritic bases (D1-D4) in E. (**G**) Peak $\Delta G/R$ of L5+ spines (S) and dendrites (D) in cell in B (blank bars; n = 13 spines) and in all cells (filled bars; n = 199 spines).

The following figure supplements are available for figure 1:

**Figure supplement 1.** Characterization and calibration of optical stimulation.

**Figure supplement 2.** Specificity of evoked presynaptic activity.

**Figure supplement 3.** Characterization of L5+ spines.

on a dendritic stretch. In total, out of 2168 analyzed spines, 199 were L5 input positive (9.2%, n = 20 cells). The average peak calcium signal amplitude in successful trials ($\Delta G/R = 0.48 \pm 0.01$) was significantly higher than the calcium signal in the parent dendrite ($\Delta G/R=0.18 \pm 0.005$, p<0.001, n = 199 spines; *Figure 1G*). There was no significant difference in the morphological properties of L5+ spines and L5 input negative (L5-) spines (length: L5+, $1.23 \pm 0.06$ μm, n = 76; L5-, $1.11 \pm 0.02$ μm, n = 419; p=0.04; normalized volume: L5+, $1.00 \pm 0.08$, n = 77; L5-, $1.00 \pm 0.03$, n = 437; p=0.97; *Figure 1—figure supplement 3A,B*).

The number of L5+ spines scaled with the total number of probed spines (*Figure 1—figure supplement 3C*) but displayed a high degree of heterogeneity between cells (4% to 29%) and individual dendritic segments (3% to 50%) as has been previously reported for Schaffer collateral synapses (*Druckmann et al., 2014*). The classical analysis of the distribution of pairwise nearest neighbor distances applied to L5+ spines (e.g. *Druckmann et al., 2014*; *Rah et al., 2013*; *Takahashi et al., 2012*) revealed significantly shorter recorded distances than expected from a random distribution (*Figure 2A*), suggesting that L5 inputs are clustered.

In order to place the clustered arrangement of synapses into the framework of the known rules of dendritic integration and arithmetic of synaptic summation (e.g. *Branco and Häusser, 2011*; *Losonczy and Magee, 2006*; *Polsky et al., 2004*), it is necessary to identify individual clusters so as to characterize for example their size and the number of inputs they receive. This means to go beyond the classical pairwise analysis of nearest neighbor distances. To this end, we developed a new approach for the identification of individual synapse clusters based on combinatorial analysis of the likelihood of specific synapse arrangements (Material and methods; *Figure 2B*).

We identified 12 clusters of L5+ spines on the basal dendrites of 20 cells (*Figure 2C* and *Figure 2—figure supplement 1*). These clusters extend over roughly 15 μm with, on average, approximately 8 spines (dendritic length, $12.52 \pm 2.40$ μm; $7.17 \pm 0.94$ L5+ spines; density, $0.84 \pm 0.19$ L5+ spines/μm; ratio of L5+ over all spines within the cluster, $0.48 \pm 0.04$; *Figure 2E,F* and *Figure 2—figure supplement 2A*; n = 12 clusters). Almost 50% of all L5+ spines were part of a cluster (86 out of 199). There is a possibility, that due to the slicing procedure some inputs may be missed. We estimate that the size of spine clusters could be underestimated by a factor between 1.4 and 2 (see Material and methods; *Figure 2—figure supplement 3*). This result shows that in the cortex cluster of synapses of an anatomically defined intracortical connection exist, which have the required size and spacing to be relevant for non-linear dendritic computations ($\geq$4 synapses within 30 μm of dendrite; see e.g. *Branco and Häusser, 2011*; *Losonczy and Magee, 2006*; *Polsky et al., 2004*). Furthermore, the scale of these clusters is in line with local learning rules (*Engert and Bonhoeffer, 1999*; *Harvey and Svoboda, 2007*), which might play a role in cluster formation and furthermore in synapse interactions within the cluster during synaptic plasticity.

## Discussion

Our results address a pressing question in the field of neural signal processing: How are the synapses of neocortical connections spatially organized on the dendrites of cortical neurons and how do the spatial patterns relate to the known rules of dendritic integration and the rules of spatiotemporal synaptic summation (e.g. *Branco and Häusser, 2011*; *Losonczy and Magee, 2006*; *Polsky et al., 2004*)?

To address this question we mapped the intralaminar synapses between L5 pyramidal neurons in visual cortex by combining optogenetics and 2-photon calcium imaging (CASIM; *Little and Carter, 2012*; *MacAskill et al., 2012*). Our results show clustering of these synapses on the basal dendrites of L5 pyramidal neurons. For quantitative analysis of the spatial organization we developed a new method for the identification of individual synapse clusters and characterization of cluster parameters.

Our results are in contrast to previous studies mapping sensory inputs in different neocortical areas, which reported no obvious spatial structure in input patterns and suggested that local non-linear dendritic integration may not be important for sensory processing in the cortex (*Chen et al., 2011*; *Jia et al., 2010*; *Varga et al., 2011*). However, more recent studies provided evidence for non-linear computations during cortical processing (*Lavzin et al., 2012*; *Smith et al., 2013*; *Xu et al., 2012*). A potential explanation for this apparent contradiction (*DeBello et al., 2014*) is that in the experiments described here, we stimulate a well-defined set of neurons, while inputs

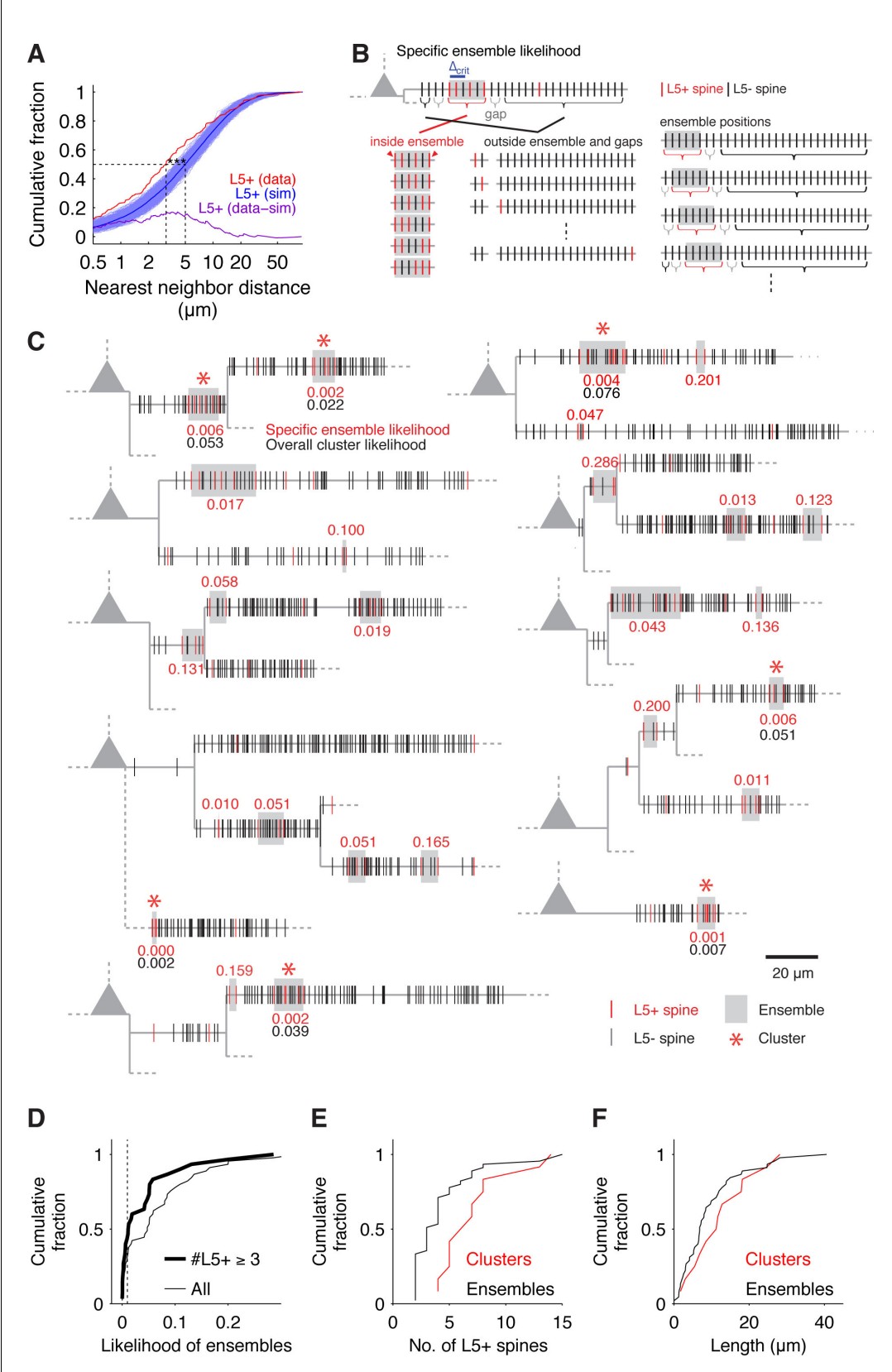

**Figure 2.** Spatial input organization. (**A**) Distribution of nearest neighbor distances between L5+ spines from experiments (red) and from 1000 times random reshuffling the recorded number of inputs over all present spines within each individual dendritic segment (blue, dark blue mean trace).
*Figure 2 continued on next page*

*Figure 2 continued*

Median values of recorded and reshuffled data are significantly different (data, median = 3.10 µm; simulations, median = 5.02 ± 0.43 µm; p<0.0001).The plot of the difference between inter-input distances from experiments and simulations (purple) shows that the deviation is most pronounced for small inter-input distances below 10 µm with a peak deviation around 5 µm, suggesting a clustered arrangement of L5+ spines. (B) Combinatorial cluster analysis: Illustration of the principle of calculating the *specific ensemble likelihood* (combinatorial cluster analysis step 2). Top, example of a dendritic segment with N = 30 spines in total and n = 5 L5+ spines (red). m = 4 L5+ spines are part of an input ensemble of size M = 6 spines (gray box) by fulfilling the maximal nearest neighbor distance criterion of $\triangle_{crit}$ = 2 (blue bar; cluster analysis step 1). The ensemble is flanked by 'gaps' of L5- spines (gray brackets). As illustrated below, the likelihood for observing an ensemble of M = 6 spines with m = 4 L5+ spines is determined by three factors: The number or ways the L5+ spines can be distributed inside the ensemble, the number of ways the remaining L5+ spines can be distributed outside of the ensemble and the number of ways an ensemble of the given size can be placed on the dendritic segment. Bottom left, all $\binom{M-2}{m-2} = 6$ possible arrangements of m = 4 L5+ spines in an ensemble of M = 6 spines, note that the ensemble edges (red arrowheads) have to be occupied always to delimit the ensemble. Bottom middle, four examples of the $\binom{N-M-2g}{n-m} = 22$ possible assignments of the remaining n-m = 1 L5+ spine to the spines outside of ensemble and gaps. Bottom right, four examples of the $N-M+1 = 25$ possibilities to place the ensemble of size M = 6 spines into the dendritic segment with N = 30 spines in total. Note that at positions close to the beginning and end of the segment, the leading or trailing gap, respectively, are not fully realized. (C) Dendrograms of 10 out of 20 mapped pyramidal cells summarizing 7 observed input clusters (L5+ spines, red; L5- spines, black; dendrograms of all other recorded cells with input clusters in *Figure 2—figure supplement 1*). Gray boxes mark input ensembles (cluster analysis step 1), red numbers are the *specific ensemble likelihood* values (cluster analysis step 2), red asterisks mark ensembles identified as clusters (*specific ensemble likelihood* ≤0.01, inputs ≥3; cluster analysis step 3), black numbers are *overall cluster likelihood* values for each cluster (cluster analysis step 4). Note that spines, which are located very close to each other, are not resolved at this scale and their marks appear to overlap; dashed lines indicate unmapped stretches). (D) *Specific ensemble likelihood* scores of all ensembles (thin line) and those with at least three inputs (thick line). Dashed line, cluster likelihood threshold. (E) Number of L5+ spines in individual clusters (red) and all ensembles (black). (F) Length of clusters (red) and all ensembles (black).

The following figure supplements are available for figure 2:

**Figure supplement 1.** Dendrograms of all other recorded cells with input clusters not shown in *Figure 2*.

**Figure supplement 2.** Combinatorial cluster analysis: Example of ensemble types and translation to non-uniform inter spine distances.

**Figure supplement 3.** Additional properties of input clusters.

**Figure supplement 4.** Comparison of the observed and expected numbers of dendritic segments containing an input cluster (cluster analysis step 5).

**Figure supplement 5.** Estimation of missed L5 inputs due to slicing.

which are defined by the same sensory stimulus (*Chen et al., 2011*; *Jia et al., 2010*; *Varga et al., 2011*) are in all likelihood arising from a mix of genetically/anatomically defined types of neurons. Input clustering has been shown for axodendritic contacts in the barn owl auditory system (*McBride et al., 2008*), and more recently, in mice, for thalamocortical (*Rah et al., 2013*) and hippocampal CA3-CA1 synapses (*Druckmann et al., 2014*). In all of these cases however there is no quantification of cluster parameters, which is crucial if one wants to relate the findings to any kind of biophysical model of dendritic computation. In this way our data and analysis go considerably beyond these reports by identifying individual clusters and quantifying cluster parameters, such as for example the number of inputs per cluster.

Our novel method for spatial cluster analysis by combinatorial statistics is not limited to the synaptic connections described here. It can be applied to characterize any distribution or pattern of synapses on the postsynaptic dendrite as obtained for example by large scale electron microscopy, GFP reconstitution across synaptic partners, array tomography or mapping of synaptic activity (e.g *Bock et al., 2011*; *Chen et al., 2011*; *Druckmann et al., 2014*; *Kleindienst et al., 2011*; *Rah et al., 2013*; *Takahashi et al., 2012*), as well as any other data regarding the spatial organization of structures along a one dimensional axis like for example the distribution of presynaptic boutons along an axon (e.g. *Schuemann et al., 2013*).

Our results show that the clusters of intracortical connections between L5 pyramidal neurons contain 4 to 14 synapses within 30 µm of basal dendrite. According to the known rules of dendritic

integration and the spatiotemporal arithmetic of synaptic summation (*Branco and Häusser, 2011*; *Losonczy and Magee, 2006*; *Polsky et al., 2004*) such clusters would result in superlinear dendritic integration during synchronous synaptic activity. Superlinear integration can also occur without tight clustering (*Branco et al., 2010*; *Losonczy and Magee, 2006*), but the specifics of the spatial input organization are expected to define how different temporal activity patterns determine the firing pattern of the postsynaptic cell. Our results are well explained by local learning rules, which may not only be involved in cluster formation but also support interactions among synapses in a cluster during synaptic plasticity events (*Engert and Bonhoeffer, 1999*; *Harvey and Svoboda, 2007*). They are also in line with observations of spatially clustered spontaneous activity of synapses during development (*Kleindienst et al., 2011*; *Takahashi et al., 2012*) and clustered spine formation and synaptic potentiation during learning and experience-dependent plasticity (*Fu et al., 2012*; *Makino and Malinow, 2011*). A further potential role of input clusters may be to overcome the stochasticity of neurotransmitter release and help to increase the reliability of synaptic transmission (*Bagnall et al., 2011*).

Typically connections between L5 pyramidal cells consist of 4 to 8 synapses, which are distributed over different branches of the postsynaptic dendrite (*Deuchars et al., 1994*; *Markram et al., 1997*). This suggests that the synapses within L5 input clusters most likely arise from a set of presynaptic neurons rather than a single one. Nevertheless, synchronous activation of L5 inputs within a cluster is expected to occur during L5 network oscillations (e.g. *Buffalo et al., 2011*; *Sun and Dan, 2009*). In anesthetized rats, L5 pyramidal cells in visual cortex receive synchronously oscillating excitatory inputs (*Sun and Dan, 2009*), which are thought to arise from rhythmically firing L5 pyramidal cells (*Flint and Connors, 1996*; *Silva et al., 1991*) within the network of L5 pyramidal cells (*Markram, 1997*). Similar oscillations in L5 have been observed in awake animals (e.g. *Buffalo et al., 2011*).

In conclusion, our results, to our knowledge, provide the first characterization of synapse clusters arising from an anatomically and genetically defined input on the dendritic tree of cortical pyramidal cells. Only such quantitative analysis of the spatial organization of inputs together with the rules of dendritic integration and synaptic summation will allow working out the logical operations of neural computations at the cellular level, which are critical for understanding information processing in neural circuits.

## Materials and methods

All experiments were conducted in compliance with the institutional guidelines of the Max Planck Society and the local government (Regierung von Oberbayern).

### Brain slice preparation

300 µm thick acute coronal slices of primary visual cortex were prepared from THY1-ChR2 (line 18) mice (*Wang et al., 2007*) between postnatal day 40 and 55 in choline-based ACSF (in mM, 110 choline chloride, 2.5 KCl, 25 $NaHCO_3$, 1.25 $NaH_2PO_4$, 0.5 $CaCl_2$, 7 $MgCl_2$, 25 D-glucose, 11.6 Na-L-ascorbate, 3.1 Na-pyruvate, aerated with 95% $O_2$/5% $CO_2$) at 0°C (*Scheuss et al., 2006*) after perfusing the animal with the same solution. Slices of the separated hemispheres were incubated at 35°C for one hour and then stored at room temperature in normal ACSF (in mM, 127 NaCl, 2.5 KCl, 25 $NaHCO_3$, 1.25 $NaH_2PO_4$, 2 $CaCl_2$, 1 $MgCl_2$, 25 D-glucose, aerated with 95% $O_2$/5% $CO_2$) (*Scheuss et al., 2006*) until the recordings.

### Photostimulus calibration and specificity testing

Cell activity was measured by cell attached recordings using patch pipettes filled with (in mM) 10 KCl, 140 K-gluconate, 10 HEPES, 2 $MgCl_2$, 2 $CaCl_2$, 0.05 Alexa 594, pH 7.25. The seal resistance was 20–40 MΩ. For obtaining ChR2 dose-response curves, light-evoked activity at blindly selected L5 pyramidal neurons was measured at increasing illumination power (1, 2, 4, 6, 9 mW at objective back aperture; see section *Imaging Experiments*) and pulse widths (2 and 5 ms) in the same ACSF solution used for mapping experiments, but additionally containing 10 µM CPP. To verify the specificity of photostimulation of L5 neurons, the activity of neurons in other cortical layers was measured in response to the same light stimulation at L5 and in the same ACSF solution as used for mapping experiments.

## Mapping experiments

Experiments were performed at room temperature in normal ACSF containing in addition (in µM) 10 NBQX, 50 picrotoxin and 10 D-serine, on a custom built two-photon laser-scanning microscope controlled with Labview (National Instruments, Austin, TX, USA) based custom imaging software. L5 pyramidal cells with undetectable or low levels of ChR2-YFP expression in primary visual cortex were visually identified (2-photon imaging at 920 nm excitation) and patched in whole-cell voltage clamp mode (access resistance 10–30 MΩ; internal solution, in mM, 125 Cs-methanesulfonate, 10 HEPES, 10 $Na_2$phosphocreatine, 4 $MgCl_2$, 4 $Na_2$-ATP, 0.4 Na-GTP, 3 Na-L-ascorbate, 5 QX-314, 10 TEA-Cl, 1 Fluo-5F, 0.03 Alexa 594, pH 7.3). Cells were filled for ≥10 min at resting potential and then depolarized to 10 mV above the NMDAR reversal potential to remove the $Mg^{2+}$ block from NMDARs (*Oertner et al., 2002*). Since in adulthood the occurrence of 'silent' synapses is effectively negligible (*Rumpel et al., 1998*) NMDA receptor mediated calcium signals are expected to identify functional synapses containing both AMPA and NMDA receptors. The photostimulus (470 nm LED; Rapp OptoElectronic GmbH, Wedel, Germany) consisted of 3 pulses of 2 ms at 30 Hz (4 mW at the objective back-aperture) and was applied via the fluorescence excitation path of the microscope to the full field of view. It evoked on average 1.57 ± 0.54 (n = 12) action potentials selectively in 58% of the ChR2 expressing L5 neurons (*Figure 1—figure supplements 1,2*). Dendritic branches were systematically scanned with overlapping rectangular tiles ($5\times19.8$ µm$^2$) at 810 nm excitation. Rotation of the tiles was adjusted to fit spines and parent dendritic segment optimally in the field of view. For calcium imaging, 50 frames (pixel size 100 x 100 nm$^2$) of single z-planes were acquired at 10 Hz in green (Fluo-5F) and red (Alexa 594) channel in the following sequence: First frame with laser shutter closed for measuring electrical offsets, 5 frames baseline before photostimulation, 2 frames with PMTs protected by a shutter during photostimulation, and 42 frames after the photostimulation. At every dendritic tile location, calcium imaging was repeated at least 3 times per z-plane (interstimulus interval, ≥10 s) at multiple z-planes to cover every spine, and an image stack was taken at higher resolution (voxel size, $50 \times 50 \times 1000$ nm$^3$).

## Input detection and structural analysis

Data were analyzed with custom software in Matlab (Mathworks, Natick, MA, USA; *Source code 1*). Spine heads and their bases on the parent dendrite were marked with rectangular ROIs, and the dendritic segments were traced in reconstructions based on averaged red channel frames from calcium imaging or the maximum z-projections of the image stacks at each tile position stitched together based on cross-correlations. Spine calcium signals ($\triangle$G/R) were calculated as change in fluorescence of the calcium indicator Fluo-5F ($\Delta$G) normalized to the mean intensity of the structural marker Alexa 594 (R) averaged over the brightest 70% of ROI pixels in the red channel. Dendrite calcium signals at the spine bases were calculated identically but without any pixel selection. $\Delta$G/R signals were smoothed (3-point moving average) and the peak amplitude in the 2nd to 4th time point after the stimulus as well as the standard deviations in the baseline interval and during the last five time points were determined. A spine was classified as receiving L5 input (L5 positive, L5+) if its peak $\Delta$G/R exceeded in one trial the dendritic peak $\Delta$G/R plus three times the higher value of the two standard deviations. We never observed large dendritic $\Delta$G/R signals at input clusters, which could lead to false negative spine categorization according to this classification, because spine/dendrite coupling is usually low (*Sabatini et al., 2002*) and simultaneous activation of many spines within a cluster was very rare. The results of the classification algorithm for spine calcium signals were verified manually. Acquisitions with high noise in the calcium signal due to low fluorescence intensity, e.g. from small or out of focus spines, and without an evident difference between the spine and the dendrite calcium signal were classified as false-positives. Acquisitions with noticeable difference in spine and dendrite calcium signal, but classified by the algorithm as no response either due to a fast decaying spine calcium signal, or because spine and the dendrite calcium signal were of similar magnitude although the spine signal peak occurred earlier, were labeled as false-negatives. Acquisitions where a subjective decision was not possible were classified as ambiguous and treated as no response. In the individual traces (n = 15293 acquisitions), the total discrepancy between manual verification and the classification by the algorithm was 2.37%, which translated into a total discrepancy of 6.46% with respect to the input classification (0.55% false negatives, 4.2% false positives, 1.71% ambiguous; n = 2168 spines). CASIM might miss synapses with very low

release probability but otherwise the false negative rate is mostly determined by the fraction of pre-synaptic neurons not expressing ChR2 as with other approaches relying on the expression of markers of any kind (e.g. *Druckmann et al., 2014*; *Rah et al., 2013*). However, since undersampling the presynaptic population is expected to be random, this would not introduce artifacts of non-random structure in the mapped synapse distribution. Positions of spines on the dendrite were extracted from 3D reconstructions of dendrite traces in time-lapse acquisitions. The location of a spine was defined to be the point on the dendrite trace with the shortest orthogonal distance to the center of the spine head.

## Cluster analysis

The combinatorial cluster analysis substantially extends the classical pairwise analysis of nearest neighbor distances (e.g. *Druckmann et al., 2014*; *Rah et al., 2013*) as it allows to identify and characterize individual clusters. The advantage of the combinatorial approach over the estimation of likelihoods using e.g. random reshuffling (e.g. (*Takahashi et al., 2012*; *Yadav et al., 2012*) is that it provides exact values for the small likelihoods involved, for which reliable estimates would require prohibitively large numbers of rounds of reshuffling. The combinatorial cluster analysis proceeds in five steps:

### Step 1: Identification of input ensembles

To identify aggregations of inputs, which represent potential clusters, we first defined input *ensembles* using as criterion the distance between nearest neighbor L5+ spines to be $\leq \triangle$crit = 10 μm (e.g. *Takahashi et al., 2012*) (*Figure 2B*, blue bar). We found 45 input ensembles in 20 cells (total analyzed dendritic length 2718 μm; *Figure 2C* and *Figure 2—figure supplement 1*, gray boxes).

### Step 2: Calculation of the *specific ensemble likelihood*

The *specific ensemble likelihood* of an ensemble is the likelihood that such ensemble of inputs occurs anywhere on the dendritic segment when the inputs are randomly assigned to the spines. It is calculated by determining the number of all possible ways of realizing such an ensemble divided by the total number of possibilities for assigning the inputs to the spines. As illustrated in *Figure 2B* for uniform inter-spine distances, the number of ways to realize a particular ensemble depends on (i) the number of ways the particular number of inputs can be assigned to the number of spines within the ensemble (*Figure 2B*, bottom, left); (ii) the number of ways the remaining inputs can be assigned to the spines outside the ensemble and its flanking gaps (*Figure 2B*, bottom, middle); and (iii) the number of possible positions of such ensemble on the dendritic segment (*Figure 2B*, bottom, right). Accounting for the gaps is necessary, because inputs to spines within the gaps would otherwise be part of the ensemble, which would then have a correspondingly larger size.

In the case of non-uniform distances an ensemble type is defined as the subset of ensembles of length $\leq l_M$, which contain $\geq m$ spines, of which $\geq m$ receive the specific input (*Figure 2—figure supplement 4B*). The number of possible input arrangements for an ensemble type is obtained by moving along the segment spine by spine and calculating the number of possible input assignments for the number of spines inside and outside of the ensemble and gaps if the ensemble definition is satisfied, which depends on the local spine density (*Figure 2—figure supplement 1B*). The likelihood that $m$ spines with the specific input (e.g. here L5 inputs) form an ensemble of size $l_M$ on a dendritic segment with $N$ spines in total and $n$ of these spines receiving the specific input in total is:

$$p_{N,n}(l_M, m, l_g) = \left( \sum_{i=1}^{N} \binom{M_i^{l_M}(m) - 2}{m - 2} \binom{N - M_i^{l_M}(m) - g_i^{l_g} - h_i^{l_g}}{n - m} \right) \frac{1}{\binom{N}{n}} \tag{1}$$

where the number of spines in an ensemble of length $l_M$ starting with the spine at position $d_i$ is

$$M_i^{l_M}(m) = \begin{cases} \text{number of spines in } [d_i, d_i + l_M] & \text{if number of spines in } [d_i, d_i + l_M] \geq m \\ 0 & \text{otherwise} \end{cases}$$

and the numbers g and h of spines in the leading and trailing gaps of length $l_g$, respectively, are

$$g_i^{l_g} = \text{number of spines in } \left[d_i - l_g, d_i\right]$$

$$h_i^{l_g} = \text{number of spines in } \left[d_i + l_M, d_i + l_M + l_g\right]$$

The first term in the sum of *Equation 1* accounts for the number of ways to assign *m*-2 inputs to an ensemble of $M_i^{l_M}(m)$-2 spines at position *i*. 2 is subtracted because inputs to the first and last spine are fixed to exclude empty ensemble edges (*Figure 2B*, bottom left; *Figure 2—figure supplement 4B*). The second term in the sum accounts for the number of ways for assigning the remaining n-m inputs to the spines outside of the ensemble and leading and trailing gap (*Figure 2B*, bottom middle). The sum runs over all spine positions on the dendritic segment (*Figure 2B*, bottom right). The denominator normalizes by the total number of possibilities for assigning n inputs to N spines. The gap length corresponds to the nearest neighbor distance criterion $l_g = \triangle_{crit}$ since an ensemble is delimited by those inputs, of which the preceding or following inputs are further away than the nearest neighbor distance criterion (*Figure 2B*, top; *Figure 2—figure supplement 4B*).

The *specific ensemble likelihood (SEL)* to observe a particular ensemble type is then the sum of *Equation 1* over all inputs available, which fit into the ensemble of size $l_M$ according to the ensemble definition above:

$$SEL_{N,n}\left(l_M, m, l_g\right) = \sum_{i=m}^{\min(n,M)} p_{N,n}\left(l_M, i, l_g\right) \tag{2}$$

The *specific ensemble likelihoods* for the observed ensembles of L5+ spines are provided in the dendrograms in *Figure 2C* and *Figure 2—figure supplement 1* (red numbers next to gray boxes).

## Step 3: Classification of ensembles as clusters

The *specific ensemble likelihood* (*Equation 2*) is then used to classify ensembles as clusters based on an empirical upper likelihood threshold of 1% (compare *Bendels et al., 2010*). The rationale behind this is that local aggregations of inputs are considered as clusters only if the probability that they would occur by chance is low. Defining ensembles of at least three inputs with a likelihood $\leq$1% as cluster, we classified 12 out of 45 ensembles as clusters (mean specific ensemble likelihood=0.003 $\pm$ 0.0008; *Figure 2C* and *Figure 2—figure supplement 1*, red asterisks). Ensembles, which fulfill the *specific cluster likelihood* threshold, but which encompass the whole dendritic segment ($d_N - d_1 \geq l_M \geq d_N - d_1 - 2 l_g$ and m = n) are not considered to qualify as a cluster.

This step is sufficient to identify clusters for their characterization. The following steps serve further confirmation of the presence of clusters and the appropriateness of the choice of the *specific ensemble likelihood* threshold.

## Step 4: Calculation of the *overall cluster likelihood* for input clusters

The fourth step is concerned with the likelihood to observe any type of cluster on a dendritic segment for comparing the numbers of expected and observed segments, which contain a cluster. This *overall cluster likelihood (OCL)* is determined by considering all possible ensemble types (see example in *Figure 2—figure supplement 4A*), calculating their *specific ensemble likelihoods (SEL, Equation 2)*, and adding up those, which are equal or below the *specific ensemble likelihood* (Step 3) of the cluster for which the *overall cluster likelihood* is calculated:

$$OCL_{N,n}\left(l_M^*, m^*, l_g\right) = \sum_{l_M} \sum_{m=2}^{\min(n,M)} SEL_{N,n}\left(l_M, m, l_g\right) \cdot \delta_{N,n}\left(l_M, m, l_g\right) \tag{3}$$

With

$$\delta_{N,n}(M, m, g) = \begin{cases} 1 & \text{if } SEL_{N,n}\left(l_M, m, l_g\right) \leq SEL_{N,n}\left(l_M^*, m^*, l_g\right) < SEL_{N,n}\left(l_M, m-1, l_g\right) \\ 0 & \text{otherwise} \end{cases}$$

where the outer sum is over all possible $l_M$ with the restriction that any set of spines, which satisfies the ensemble definitions for more than one $l_M$ is only counted once. The *overall cluster likelihood* for

the observed clusters of L5+ spines are provided in the dendrograms in *Figure 2C* and *Figure 2—figure supplement 1* (black numbers below gray boxes with red asterisks).

## Step 5: Comparison of the observed and expected numbers of dendritic segments containing an input cluster

In the final step, the *overall cluster likelihood* is used to calculate the probability of finding a cluster in 12 out of 40 analyzed segments. Since the *overall cluster likelihood (OCL)* is different in different segments, we sorted its values in ascending order (*Figure 2—figure supplement 5A*). The probability to obtain at least *c* segments containing a cluster with an of $OCL \leq OCL_{max}(c)$ in the data set of S segments (in the present case 40) is calculated by binomial statistics from the *maximal overall cluster likelihood* $OCL_{max}(c)$ as upper bound on OCL:

$$P(c, OCL_{max}) = \sum_{x=c}^{S} \binom{S}{x} OCL_{max}^{x} (1 - OCL_{max})^{S-x} \qquad (4)$$

For this analysis only dendritic segments with a length of $\geq 20$ μm were included. This probability is equivalent to the P value of the binomial test for the null hypothesis that the number of observed segments with an input cluster arises from a random input distribution (*Yadav et al., 2012*). We find that the probability to observe c segments containing a cluster in the data set of 40 segments in total declines and remains below the significance threshold of 0.001 until the second last of the sorted segments (*Figure 2—figure supplement 5B*). The last segment with a cluster has a comparatively high OCL value (*Figure 2—figure supplement 5A*), which drives the probability to observe the given number of segments with a cluster above the significance threshold. However, the number of 11 segments with a cluster with a P value below 0.001 supports the hypothesis that L5+ spines are indeed arranged in clusters not by chance. This step serves further as confirmation for the appropriate choice of the cluster criterion in step 3.

## Estimation of missed L5 inputs due to slicing

To estimate the fraction of missed L5 inputs resulting from the slicing procedure we calculated the radial axon density *A(r)* (*Figure 2—figure supplement 3B*) of L5 pyramidal cell axons in mouse visual cortex based on the Sholl analysis by *Blackman et al., 2014* (their *Figure 2C*, 2nd panel from left) as

$$A(r) = \frac{number\ of\ crossings\ at\ r}{4\pi r^2 \delta r} \qquad (5)$$

The average axon density arising from presynaptic L5 neurons at a distance *r* from a dendritic location is given by the radial axon density of L5 pyramidal cells at the distance of *r* from the soma. For calculating the remaining fraction of presynaptic neurons contributing to the axon density at a dendritic location at a depth of D in a slice of thickness T, three cases have to be distinguished (*Figure 2—figure supplement 3A*). The neuron density is assumed to be uniform and therefore needs not to be explicitly considered in the following equations:

Case 1: Spherical shell around dendritic location in slice is complete, 0 < r < D (*Figure 2—figure supplement 3A*, top panel)

The volume of a spherical shell of radius *r* and thickness $\delta r$ is

$$V(r) = 4\pi r^2 \delta r \qquad (6)$$

Case 2: Spherical shell around dendritic location in slice with one cap missing, D < r < T-D (*Figure 2—figure supplement 3A*, middle panel)

The volume of a spherical shell of radius *r* and thickness $\delta r$ with one cap of height *h = r-D* subtracted is

$$V(r) = \left(4\pi r^2 - 2\pi rh\right)\delta r = 2\pi r(r + D)\delta r \qquad (7)$$

Case 3: Spherical shell around dendritic location in slice with two caps missing, T-D < r < $r_{max}$ (*Figure 2—figure supplement 3A*, bottom panel)

The maximal radius $r_{max}$ denotes here the maximal radial extent of the average axonal density. The volume of a spherical shell of radius $r$ and thickness $\delta r$ with top cap of height $h_1 = r-D$ and bottom cap of height $h_2 = r-(T-D)$ subtracted is

$$V(r) = \left(4\pi r^2 - 2\pi r h_1 - 2\pi r h_2\right)\delta r = 2\pi r T \delta r \tag{8}$$

Combining these equations yields the remaining fraction of axonal density within a slice of thickness T as a function of the depth D in the slice as

$$F(D) = \frac{\int_0^D 4\pi r^2 A(r)dr \ + \ \int_D^{T-D} 2\pi r(r+D)A(r)dr + \ \int_{T-D}^{r_{max}} 2\pi r T A(r)dr}{\int_0^{r_{max}} 4\pi r^2 A(r)dr} \tag{9}$$

From this we estimate that the fraction of the remaining L5 pyramidal cell axon density that can be activated in a slice is between 50% and 70% depending on the depth in the slice (*Figure 2—figure supplement 3C*). This suggests that between 30% and 50% of L5 inputs are missed due to the slicing process.

## Statistics

Spine and dendrite calcium signal peaks were compared to each other by Wilcoxon rank sum test. Z-tests were applied when comparing the median inter-input distance from the experiments to Monte Carlo simulations. In plots asterisks indicate p-values as *, $p<0.05$; **, $p<0.01$; and ***, $p<0.001$.

The values given in the text are mean ± SEM (standard error of the mean), if not indicated otherwise.

## Acknowledgements

We thank Volker Staiger, Claudia Huber, Frank Voss, Helena Tultschin and Max Sperling for their technical assistance and Christian Leibold, Christian Lohmann and Moritz Helmstaedter for comments on an earlier version of the manuscript. This work was supported by the Max Planck Society and the Human Frontier Science Program (VS).

## Additional information

### Funding

| Funder | Grant reference number | Author |
|---|---|---|
| Max-Planck-Gesellschaft | | Onur Gökçe<br>Tobias Bonhoeffer<br>Volker Scheuss |
| Deutsche Forschungsgemeinschaft | SFB 870 | Tobias Bonhoeffer<br>Volker Scheuss |
| Human Frontier Science Program | Career Development Award - 00035/2009 | Volker Scheuss |

The funders had no role in study design, data collection and interpretation, or the decision to submit the work for publication.

### Author contributions

OG, Acquisition of data, Analysis and interpretation of data, Drafting or revising the article; TB, Conception and design, Analysis and interpretation of data, Drafting or revising the article; VS, Conception and design, Acquisition of data, Analysis and interpretation of data, Drafting or revising the article

### Author ORCIDs

Onur Gökçe, http://orcid.org/0000-0002-3913-3893
Tobias Bonhoeffer, http://orcid.org/0000-0001-7897-6634
Volker Scheuss, http://orcid.org/0000-0002-0659-3228

## Ethics

Animal experimentation: All experiments were conducted in compliance with the institutional guidelines of the Max Planck Society and the local government (Regierung von Oberbayern). The presented work involved experiments on brain tissue obtained after painless killing. The number of animals was reported to the authorities as required. No ethical approval reference number is being issued for such experiments.

## Additional files

### Supplementary files

• Source code 1. Custom software in Matlab for data analysis.

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
