## [Decision Letter]

Thank you for submitting your work entitled "Clusters of synaptic inputs on dendrites of layer 5 pyramidal cells in visual cortex" for peer review at *eLife*. Your submission has been favorably evaluated by Eve Marder (Senior editor), a Reviewing Editor, and two reviewers.

The reviewers have discussed the reviews with one another and the Reviewing Editor has drafted this decision to help you prepare a revised submission.

Summary:

This study maps excitatory inputs onto dendrites of layer 5 pyramidal cells in visual cortex with optogenetic tools to study whether or not activated spines can form stable input clusters over time. This is an important and interesting topic, since synaptic input clustering has a large effect on the mode of input integration and plasticity in cortical pyramidal cells. Indeed, a number of papers have been published on this issue recently pro and contra (e.g. Kleindienst et al., Neuron, 2011; Takahashi et al., Science, 2012; Chen et al., 2011, Nature). Specifically, the authors investigated how spines in close proximity respond when a large population of ChR2+ layer 5 axons are activated simultaneously by developing a novel, optogenetics-based method. They conclude that L5-L5 synapses tend to cluster on the same dendritic branch (within a dendritic stretch of 20 um) in clusters containing 4 to 8 activated spines. The manuscript thus makes a valuable contribution to an ongoing debate in the field. However, several issues need to be addressed.

Essential revisions:

1) The authors identified 7 L5-L5 clusters (out of 30 ensembles) on dendrites arising from 13 cells (Figure 2). While statistically significant, these are rather small numbers and, thus, are insufficient to adequately quantify the real statistics of L5 clusters (i.e., are there really 4-8 activated spines on such a cluster?). As the authors argue – justifiably – that the main advantage of their method is to identify functional connections, the manuscript would be strengthened by improving the statistics, with more convincing quantification of the cluster size.

2) The authors leave open the most important question: how do these simultaneously active spines contribute to the activation of the branch segment? Are they more effective in evoking supralinear responses opposed to randomly activated spines? Indeed, the authors suggest (based on other studies) that, in L5-L5 clusters, there would be non-linear dendritic integration of synaptic inputs during synchronous activation. But they do activate L5 synchronously with optogenetics, so they should observe this nonlinearity (e.g. local NMDA spikes – in the experiment suggested below – and/or large Ca signals in both the corresponding dendritic spines and dendritic shaft, belonging to the cluster – as opposed to cases where L5 synapses are not clustered). The authors seem to be in an ideal position to examine whether their morphological clusters are indeed operating in the supra-linear regime. This could be examined as follows: after identifying a cluster, one could hyperpolarize the soma of the post-synaptic neuron (to avoid axonal spiking) and compare the local dendritic/spine calcium signals in a cluster vs. non-cluster cases.

3) The authors tried to pharmacologically exclude AMPA type receptors from their analysis. However, these receptors may also contribute to the activation of dendritic branches even though the calcium transients cannot be detected. The point here is that throughout the experiments the authors blocked AMPA receptors under the assumption that all synapses contain both NMDARs and AMPARs, so the underlying population they sample from would not change. To our knowledge, this assumption has not been directly tested for synapses between L5 PCs (i.e. lack of silent – NMDA only synapses), and the citation the authors provide on this issue deals with silent synapses between L2/3 PCs or from L2/3 to L5 synapses.

4) What% of excitatory inputs onto L5PCs is missing due to the slicing process and how much this affects the authors' estimate of the size of the active spine clusters?

5) In their Discussion, the authors should address the question of synchrony. Namely, how likely is it that the L5 population is activated synchronously for visual (rather than optogenetic) input? (In this context it is worth mentioning that individual L5-L5 synapses consists of 4-5 connections that are typically formed on different dendritic branches. These 4-5 connections will be activated simultaneously but, obviously, they are not part of a cluster).

---

## [Author Response]

1) The authors identified 7 L5-L5 clusters (out of 30 ensembles) on dendrites arising from 13 cells (Figure 2). While statistically significant, these are rather small numbers and, thus, are insufficient to adequately quantify the real statistics of L5 clusters (i.e., are there really 4-8 activated spines on such a cluster?). As the authors argue – justifiably – that the main advantage of their method is to identify functional connections, the manuscript would be strengthened by improving the statistics, with more convincing quantification of the cluster size.

We performed additional mapping experiments on 7 cells. In total we now mapped 20 cells and found 12 clusters. The updated cluster parameters are as follows:

“These clusters extend over roughly 15 μm with, on average, approximately 8 spines (dendritic length, 12.52 ± 2.40 μm; 7.17 ± 0.94L5+ spines; density, 0.84 ± 0.19 L5+ spines/μm; ratio of L5+ over all spines within the cluster, 0.48 ± 0.04; Figure 2 and Figure 2—figure supplement 2; n = 12 clusters). Almost 50% of all L5+ spines were part of a cluster (86 out of 199).”

We updated the respective plots in Figure 1 (panel G), Figure 2 (panels A, D, E, F), Figure 1—figure supplement 3 (panels A-D), now Figure 2—figure supplement 2 (panels A-D).

In order to provide a dataset which is as complete as possible we have decided to display the dendrograms of the additionally mapped cells in the supplemental data. They are shown in the new Figure 2—figure supplement 1.

Furthermore, the new data prompted us to refine Step 5 of the subsection “Cluster analysis”; Figure 2—figure supplement 5).

2) The authors leave open the most important question: how do these simultaneously active spines contribute to the activation of the branch segment? Are they more effective in evoking supralinear responses opposed to randomly activated spines? Indeed, the authors suggest (based on other studies) that, in L5-L5 clusters, there would be non-linear dendritic integration of synaptic inputs during synchronous activation. But they do activate L5 synchronously with optogenetics, so they should observe this nonlinearity (e.g. local NMDA spikes – in the experiment suggested below – and/or large Ca signals in both the corresponding dendritic spines and dendritic shaft, belonging to the cluster – as opposed to cases where L5 synapses are not clustered). The authors seem to be in an ideal position to examine whether their morphological clusters are indeed operating in the supra-linear regime. This could be examined as follows: after identifying a cluster, one could hyperpolarize the soma of the post-synaptic neuron (to avoid axonal spiking) and compare the local dendritic/spine calcium signals in a cluster vs. non-cluster cases.

This is an interesting suggestion. Note, however, that the current mapping approach involves blocking AMPARs with NBQX (compare point 3) to reduce synaptic transmission in order to prevent unspecific spread of excitation to other neurons. Blocking AMPARs is incompatible with the suggested experiment.

However, in our experiments we never knew for sure that unspecific excitation would be a problem. We just assumed it would be and blocked AMPARs preemptively.

Therefore, we tested if optogenetic stimulation remains specific and contained to L5 pyramidal cells even without blocking AMPARs. This experiment showed that optogenetic stimulation is not specific and contained to L5 pyramidal cells without blocking AMPAR. In other words the feared unspecific spread of excitation indeed occurred. 6 out of 15 probed L2/3 neurons show APs in response to optogenetic stimulation in L5 (see Figure 3, analogous to Figure 1—figure supplement 2 “Specificity of Evoked Presynaptic Activity”). We therefore have to conclude that the dendritic integration experiment cannot be performed as suggested.

Author response image 1.Analogous to data in Figure 1—figure supplement 2: “Specificity of Evoked Presynaptic Activity” but here without application of NBQX.AP generation in neurons in L2/3 measured by cell attached recordings during a wide-field light stimulus in L5 (green circle). Left panel: marks (+) indicate the vertical location of the cells. Color codes for the neurons with no evoked APs (blue) and evoked APs (red). Right panel: example traces.**DOI:**
http://dx.doi.org/10.7554/eLife.09222.014

3) The authors tried to pharmacologically exclude AMPA type receptors from their analysis. However, these receptors may also contribute to the activation of dendritic branches even though the calcium transients cannot be detected. The point here is that throughout the experiments the authors blocked AMPA receptors under the assumption that all synapses contain both NMDARs and AMPARs, so the underlying population they sample from would not change. To our knowledge, this assumption has not been directly tested for synapses between L5 PCs (i.e. lack of silent – NMDA only synapses), and the citation the authors provide on this issue deals with silent synapses between L2/3 PCs or from L2/3 to L5 synapses.

To address this question we performed an additional experiment series with double patch clamp recordings from L5 pyramidal cells in order to compare the failure rates of postsynaptic responses at -70 mV (AMPAR EPSCS) and +40 mV (NMDAR EPSCS) holding potential. In the absence of silent synapses the failure rates should be equal. In patch clamp recordings that we performed on 40 pairs of L5 pyramidal cells we did not find a single connection. We had expected a connection probability of ~10% (e.g. Markram et al., J Physiol 1997; Thomson et al., Cereb Cortex 2002; Brown & Hestrin, Nature 2009). A possible reason for the apparent difference is that the value of ~10% is not from adult mouse visual cortex, but from young/adult rat sensory cortex and young mouse visual cortex. A recent publication confirms our result (Jiang X et al., Science, vol. 350, issue 6264, 2015; e.g. their Figure 4A). This however also means that we cannot address the question in the way that we had planned.

However, Rumpel et al. (J Neurosci, vol. 18, issue 21, 1998) have previously shown that there are no silent synapses within L5 and L6 of maturing rat visual cortex. This study used fiber stimulation in L5, which does not exclusively activate L5 inputs to L5 neurons, but the general absence of silent synapses also suggests the absence of silent synapses between L5 neurons. Therefore, we replaced the references on silent synapses by the reference to Rumpel et al. in the subsection “Mapping experiments”.

4) What% of excitatory inputs onto L5PCs is missing due to the slicing process and how much this affects the authors' estimate of the size of the active spine clusters?

We used the published Sholl analysis of axonal arbors of L5 pyramidal cells in mouse visual cortex by Blackman et al. (Frontiers in Neuroanatomy, Volume 8, Article 65, 2014; their Figure 2C, 2nd panel from left) to estimate the percentage of missed L5 inputs by slicing. We estimate that the fraction of the remaining axon density that can be activated in a slice is between 50% and 70% depending on the depth in the slice. Thus between 30% and 50% of inputs are missed due to the slicing process such that the size of spine clusters might be larger by a factor between 1.4 and 2 than the measured value.

We added this information in the text: “There is a possibility, that due to the slicing procedure some inputs may be missed. We estimate that the size of spine clusters could be underestimated by a factor between 1.4 and 2 (see Materials and methods; Figure 2—figure supplement 3).”

We added the equations for estimating the fraction of missing contacts as the Materials and methods section *“*Estimation of L5 inputs due to slicing”:

“To estimate the fraction of missed L5 inputs resulting from the slicing procedure we calculated the radial axon density A(r) (Figure 2—figure supplement 3) of L5 pyramidal cell axons in mouse visual cortex based on the Sholl analysis by Blackman et al. 2014 (Figure 2, 2nd panel from left) as

(5) A(r)=number of crossings at r4πr2δr […] From this we estimate that the fraction of the remaining L5 pyramidal cell axon density that can be activated in a slice is between 50% and 70% depending on the depth in the slice (Figure 2—figure supplement 3). This suggests that between 30% and 50% of L5 inputs are missed due to the slicing process.”

We added a supplemental Figure “Figure—figure supplement 5: Estimation of missed L5 inputs due to slicing”that illustrates the approach and the estimated fraction of the remaining axon density in a slice.

*5) In their Discussion, the authors should address the question of synchrony. Namely, how likely is it that the L5 population is activated synchronously for visual (rather than optogenetic) input? (In this context it is worth mentioning that individual L5-L5 synapses consists of 4-5 connections that are typically formed on different dendritic branches. These 4-5 connections will be activated simultaneously but, obviously, they are not part of a cluster).*

We now added this point to the Discussion, and also discuss how synchronous activation of the synapses within a cluster is expected to arise during L5 network oscillations:

“Typically connections between L5 pyramidal cells consist of 4 to 8 synapses, which are distributed over different branches of the postsynaptic dendrite (Deuchars et al., 1994; Markram et al., 1997). […] Similar oscillations in L5 have been observed in awake animals (e.g. Buffalo et al., 2011).”